# Epigenetic Echoes: Bridging Nature, Nurture, and Healing Across Generations

**DOI:** 10.3390/ijms26073075

**Published:** 2025-03-27

**Authors:** Blerida Banushi, Jemma Collova, Helen Milroy

**Affiliations:** School of Indigenous Studies, The University of Western Australia, Crawley, WA 6009, Australia; jemma.collova@uwa.edu.au (J.C.); helen.milroy@uwa.edu.au (H.M.)

**Keywords:** trauma, intergenerational, transgenerational, epigenetics, DNA methylation, non-coding RNA, stress, inheritance, histones, neuroplasticity, psychedelics

## Abstract

Trauma can impact individuals within a generation (intragenerational) and future generations (transgenerational) through a complex interplay of biological and environmental factors. This review explores the epigenetic mechanisms that have been correlated with the effects of trauma across generations, including DNA methylation, histone modifications, and non-coding RNAs. These mechanisms can regulate the expression of stress-related genes (such as the glucocorticoid receptor (*NR3C1*) and FK506 binding protein 5 (*FKBP5*) gene), linking trauma to biological pathways that may affect long-term stress regulation and health outcomes. Although research using model organisms has elucidated potential epigenetic mechanisms underlying the intergenerational effects of trauma, applying these findings to human populations remains challenging due to confounding variables, methodological limitations, and ethical considerations. This complexity is compounded by difficulties in establishing causality and in disentangling epigenetic influences from shared environmental factors. Emerging therapies, such as psychedelic-assisted treatments and mind–body interventions, offer promising avenues to address both the psychological and potential epigenetic aspects of trauma. However, translating these findings into effective interventions will require interdisciplinary methods and culturally sensitive approaches. Enriched environments, cultural reconnection, and psychosocial interventions have shown the potential to mitigate trauma’s impacts within and across generations. By integrating biological, social, and cultural perspectives, this review highlights the critical importance of interdisciplinary frameworks in breaking cycles of trauma, fostering resilience, and advancing comprehensive healing across generations.

## 1. Introduction

Epigenetics, first proposed by Conrad Hal Waddington in the 1940s [1], refers to stable but reversible changes in gene expression that occur without alterations to the primary DNA sequence [2,3]. The definition has since expanded to include changes in gene expression that are not necessarily heritable and may not always be reversible [4]. These changes are mediated by several key molecular mechanisms, including DNA methylation, hydroxymethylation, histone post-translational modifications, nucleosome positioning and histone variants, 3D genome organization, and the regulation of gene activity by non-coding RNAs (ncRNAs) (Figure 1) [5,6]. Collectively, these mechanisms shape the epigenome, a genome-wide landscape of DNA and chromatin modifications that regulates gene expression in response to environmental stimuli and developmental signals [7]. 

Among epigenetic mechanisms, DNA methylation has been one of the most extensively studied [8,9] (Figure 1). DNA methylation involves the covalent addition of a methyl group to the 5’ position of cytosine residues within CpG dinucleotides, forming 5-methylcytosine (5mC) [2,8]. This modification is generally associated with stable, heritable gene silencing and plays a critical role in key biological processes, including gene regulation, embryonic development, cellular differentiation, genomic imprinting, transposable element suppression, and X-chromosome inactivation [9,10]. However, in specific genomic or developmental contexts, DNA methylation can also be linked to gene activation [11,12]. Despite its relative stability, 5mC can be dynamically reversed through passive dilution during DNA replication or via active enzymatic demethylation [13,14]. The latter process involves ten-eleven translocation (TET) enzymes, which oxidize 5mC to 5-hydroxymethylcytosine (5hmC) and further oxidation products that facilitate cytosine demethylation [14,15,16]. Notably, 5hmC is now recognized as an independent epigenetic mark with distinct roles in gene regulation and cellular differentiation [17,18].

In the male germline, extensive chromatin reorganization occurs during spermiogenesis: histones are largely replaced by protamines to hyper-condense sperm chromatin and ensure genomic integrity [19]. Disruptions in histone-to-protamine replacement are associated with abnormal sperm function and infertility [19,20]. Both histones and protamines can carry a variety of post-translational modifications (PTMs)—including acetylation, methylation, phosphorylation, and ubiquitination, as well as less common marks, like malonylation, crotonylation, and lactylation—that are crucial for chromatin organization and gene regulation [2,21,22,23]. By altering DNA accessibility and recruiting specific chromatin-associated proteins, these PTMs contribute to a combinatorial “histone code”, whereby particular modification patterns influence transcription, DNA repair, replication, and differentiation [21,22,24,25,26]. While the histone code hypothesis has guided the understanding of histone PTMs, emerging evidence reveals additional regulatory crosstalk between histone modifications, DNA methylation, and chromatin-binding proteins [21,24,27,28].

Non-coding RNAs, which do not encode proteins, play pivotal roles in epigenetic regulation by modulating gene expression at the transcriptional, post-transcriptional, and chromatin remodeling levels [29] (Figure 1). They are essential for normal development and contribute to the pathogenesis of various diseases [30,31,32]. Key categories of ncRNAs include microRNAs (miRNAs), which typically bind complementary sequences in target mRNAs to promote their degradation or inhibit translation, and long non-coding RNAs (lncRNAs), which can regulate chromatin architecture and gene transcription by serving as scaffolds for chromatin-modifying complexes [30,32]. Circular RNAs (circRNAs), initially discovered as miRNA “sponges,” also interact with RNA-binding proteins, regulate transcription, and, in some cases, encode functional micropeptides [33,34].

The interplay between PTMs, DNA methylation, and ncRNAs forms a complex regulatory network that regulates gene expression in a context-dependent manner [35,36]. For example, certain lncRNAs recruit histone-modifying complexes like Polycomb Repressive Complex 2 to specific loci, inducing repressive histone marks (e.g., H3K27me3) and silencing gene expression [35,36,37]. Similarly, DNA methylation can influence the expression of ncRNAs, which in turn regulate chromatin state by guiding histone-modifying enzymes or DNA methyltransferases [35,36]. This interconnected system enables cells to adapt to environmental and developmental signals while still preserving epigenetic memory across cell divisions [21,38,39].

Throughout life, the epigenome undergoes dynamic reprogramming. During early embryogenesis and primordial germ cell development, epigenetic marks are erased and reset to establish totipotency and pluripotency, while in adult tissues, epigenetic marks are continuously remodeled in response to environmental influences and life experiences while maintaining cell type-specific gene expression patterns [40,41,42,43]. For example, in immune cells, DNA methylation and histone modifications are reconfigured during inflammation or infection, enabling rapid changes in gene expression [44,45]. In neural tissues, changes in histone acetylation and DNA hydroxymethylation are critical for learning, memory formation, and synaptic plasticity [46,47]. These properties underscore the epigenome’s responsiveness to environmental stimuli across the lifespan.

A growing body of research over the past two decades has postulated that traumatic experiences, exposure to environmental toxins, or nutritional deficiencies can induce epigenetic changes that, in some cases, might be transmitted to offspring [48,49,50,51,52]. While epigenetic inheritance is well-established in plants, nematodes, and *D. melanogaster*, its extent in mammals, particularly humans, remains under investigation and is a subject of active debate [53,54,55,56,57].

When discussing epigenetic inheritance, it is important to distinguish between intergenerational and transgenerational inheritance (Figure 2) [57,58].

Intergenerational epigenetic effects are those observed in the immediate offspring (the F1 generation) of an exposed individual (the F0 generation) [59]. In mammals, for maternal exposures during pregnancy, intergenerational effects can extend to the F2 generation because the developing fetus (F1) and its in utero germ cells (the future F2) are directly exposed to the environmental factor [58,59,60]. In contrast, transgenerational epigenetic inheritance refers to effects that manifest in generations that were not directly exposed to the original environmental factor (e.g., in F2 or later, for paternal exposures, and F3 or later, for maternal exposures) (Figure 2) [58,59,60]. For example, maternal smoking during pregnancy represents an intergenerational exposure: it has been associated with altered DNA methylation at specific loci in newborns (F1), such as the *AHRR* (Aryl-Hydrocarbon Receptor Repressor) and *GFI1* (Growth Factor Independent 1) genes, reflecting in utero tobacco exposure and its impact on pathways related to growth and immune function [61,62]. To be considered transgenerational, an epigenetic modification induced by an exposure must be transmitted through the germline and detected in offspring generations that had no direct exposure [58,59,60]. In mammals, this means transgenerational inheritance requires that an epigenetic change escapes the comprehensive epigenetic reprogramming that occurs in germ cells and early embryos.

Initial human studies have reported associations between extreme ancestral experiences and epigenetic marks in descendants [57,63]. For instance, research on adult offspring of Holocaust survivors has reported differential DNA methylation patterns in stress-regulatory genes compared to control populations [64,65]. Similarly, the prenatal exposure of mothers to the Dutch Hunger Winter famine of 1944–45 has been linked to persistent DNA methylation changes in their offspring decades later [66,67]. These pioneering studies suggest that severe trauma or nutritional stress in one generation may correlate with biological changes in the next. However, it is crucial to note that such human data are largely correlational and do not establish causation [57,63]. Many confounding factors, ranging from parenting behaviors and socioeconomic conditions to continued exposure to community-level trauma, could produce similar outcomes, making it challenging to pinpoint epigenetics as the direct cause [57,68,69].

This review, through an integrative perspective, emphasizes both the promise and the limitations of the current research and calls for interdisciplinary, ethically sound methodologies capable of advancing our understanding and treatment of generational trauma.

## 2. DNA Methylation and Trauma Across Generations

DNA methylation is a key epigenetic mechanism involved in regulating biological responses to environmental stimuli. Evidence from animal models indicates that environmentally induced DNA methylation modifications, in some cases, could be transmitted to the next generation. However, in humans, the evidence remains largely inconclusive, with most findings being correlational rather than demonstrating direct causality [56,65,70,71]. For example, decreased methylation of the *FKBP5* gene (which encodes FK506-binding protein 5, a key regulator of the stress response) has been observed in Holocaust descendants [64,65]. This hypomethylation is associated with dysregulation of the hypothalamic–pituitary–adrenal (HPA) axis, including abnormal cortisol secretion and reduced glucocorticoid receptor sensitivity, which correlates with increased susceptibility to post-traumatic stress disorder (PTSD) and anxiety disorders in offspring [64,65,72,73,74]. It must be noted, however, that the link between *FKBP5* methylation and PTSD risk is complex and not deterministic. While some studies report *FKBP5* hypomethylation in individuals with PTSD [75], others have found increased methylation at certain *FKBP5* loci depending on tissue type, trauma exposure timing, and genetic background [76]. Interestingly, holocaust exposure had an effect on *FKBP5* methylation observed in both survivors and their offspring, albeit in opposite directions (survivors showed elevated methylation at one locus, while offspring showed lower methylation) [64]. Both changes were correlated between parent and child, suggesting a link, though the mechanism remains unclear [64].

Interpreting these human findings is further complicated by shared environmental influences. For instance, in the case of Holocaust survivor families, parental PTSD symptoms and parenting behaviors could also affect offspring cortisol levels, stress reactivity, and anxiety, independent of any inherited epigenetic changes [64]. Cultural narratives and the collective family experience of trauma can shape a descendant’s coping mechanisms and stress responses, adding another layer of complexity to the interpretation of epigenetic differences [64].

Prenatal exposure to maternal stress represents another context in which DNA methylation changes have been documented. Maternal stress during pregnancy has been linked to altered DNA methylation of the glucocorticoid receptor gene (*NR3C1*) promoter in offspring, which in turn is associated with dysregulation of the HPA axis, heightened stress sensitivity, and elevated risk of neuropsychiatric disorders in those descendants [77]. For example, in one study, newborns of mothers who experienced depression during pregnancy showed increased *NR3C1* methylation, which correlated with heightened cortisol stress responses in the infants [78]. Similarly, animal studies have shown that prenatal exposure to stress in rodents is associated with increased *NR3C1* methylation in offspring, which, in turn, modulates their stress reactivity [79]. The magnitude and persistence of such prenatal stress-induced methylation changes depend on factors like the timing, severity, and nature of the stressor [77]. Moreover, postnatal environmental factors, especially maternal care, play a critical role in modulating these effects. In rodents, high-quality maternal care (e.g., frequent licking/grooming of pups) can reverse or mitigate the adverse epigenetic and physiological effects of prenatal stress, thereby normalizing HPA axis function and stress responses in the offspring [80]. In humans, considerable variability in *NR3C1* methylation and its influence on stress outcomes suggests that these effects are highly context-dependent; genetic predispositions, the postnatal rearing environment, and individual differences all contribute to whether a prenatal stress exposure translates into lasting epigenetic and phenotypic changes [81].

At a mechanistic level, elevated maternal glucocorticoid levels during stress can cross the placenta, affecting fetal epigenetic programming. Normally, the placenta expresses the enzyme 11β-hydroxysteroid dehydrogenase type 2 (11β-HSD2), which converts active cortisol to its inactive form cortisone, protecting the fetus from excessive glucocorticoid exposure [82]. Prenatal stress can reduce placental 11β-HSD2 activity, diminishing this protective barrier and leading to heightened fetal exposure to maternal cortisol [83]. Decreased placental 11β-HSD2 has been correlated with lower birth weight and higher blood pressure in later life [84], consistent with the developmental origins of health and disease hypothesis [85]. Indeed, maternal stress and anxiety are associated with reduced placental 11β-HSD2 expression in humans [83], linking maternal mood directly to the hormonal environment of the fetus.

Research on populations that have experienced severe trauma offers further evidence of DNA methylation changes correlated with intergenerational effects. For example, a pilot study of mother–child dyads who survived the Rwandan genocide found increased DNA methylation at several stress-related genes—including *BCOR* (BCL6 Corepressor), *PRDM8* (PR/SET Domain 8), and *VWDE* (Von Willebrand Factor D and EGF Domains)—in both the exposed mothers and their children [86,87]. These genes are involved in transcriptional regulation and developmental pathways, and hypermethylation would typically suggest reduced gene expression. Interestingly, there is evidence that blood DNA methylation patterns can reflect those in brain tissue for certain stress-responsive genes, suggesting that peripheral methylation changes might parallel central nervous system changes related to emotion and memory [86,87]. Nonetheless, the functional significance of these methylation changes across generations in humans remains speculative, likely shaped by additional factors, such as maternal caregiving behaviors and shared environmental stressors [70,86,87,88,89,90,91]. This highlights the difficulty in disentangling inherited epigenetic modifications from environmental and cultural influences—which we explored in later sections of this review.

In summary, DNA methylation changes are among the most documented epigenetic alterations in studies of trauma and stress across generations. They provide a plausible molecular link between environmental experiences (such as severe stress) and changes in gene regulation in offspring. Nevertheless, current evidence, especially in humans, is associative; rigorous longitudinal and mechanistic studies are needed to determine which methylation changes are truly causal in mediating outcomes and which are simply correlates or biomarkers of trauma exposure. DNA methylation marks operate within a broader epigenetic context that includes chromatin state and ncRNA regulation, and thus, they must be interpreted within the network of other molecular changes and environmental factors at play.

## 3. Histone Modifications

Histone post-translational modifications (e.g., methylation and acetylation) are central regulators of chromatin structure and gene expression, and they can be dynamically altered by environmental stimuli [71,92]. For example, trimethylation of histone H3 at lysine 27 (H3K27me3) by Polycomb repressive complex 2 is associated with chromatin compaction and gene silencing, whereas acetylation of H3 lysine 9 (H3K9ac) by histone acetyltransferases (HATs) is linked to an open chromatin state and active transcription [71,92,93]. These modifications serve as molecular switches between heterochromatin (condensed, inactive) and euchromatin (relaxed, active), enabling cells to fine-tune gene expression in response to stress and other external signals. Notably, in model organisms, stress can modulate histone-modifying enzymes, leading to changes in histone marks at stress-responsive genes [71,94,95]. Some well-characterized effects of chronic stress include increased H3 phosphoacetylation and altered H3K9/H3K27 methylation at loci regulating the HPA axis (e.g., *CRH* and *GR* genes) and neurotrophic factors such as the Brain-Derived Neurotrophic Factor (*BDNF*) [94]. Through such mechanisms, environmental stressors like trauma can induce biochemical modifications to the chromatin structure, resulting in altered patterns of gene expression. One illustrative example is the effect of early-life maternal care on histone acetylation in rodent models. High levels of maternal licking and grooming enhance histone H3 acetylation at the glucocorticoid receptor *GR* gene promoter in the hippocampus of offspring [80,96]. This acetylation is mediated by the CREB-binding protein (CBP/p300) HAT complex, which relaxes chromatin and elevates *GR* gene transcription [80,96]. Moreover, maternal nurturing induces a persistent reduction in histone deacetylase (HDAC) activity at the *GR* promoter, helping to maintain an open chromatin state and sustain *GR* expression [80,96]. These findings underscore some mechanistic basis in rodent models by which early-life environmental factors can induce lasting epigenetic changes that influence stress reactivity and behavioral outcomes.

Similarly, in model organisms, paternal environmental exposures, such as diet and stress, can induce epigenetic changes in sperm, particularly through changes in histone modifications, with potential consequences for offspring metabolism [97,98,99]. For example, studies in mice have shown that a high-fat diet reduces H3K27me3 levels at key loci regulating metabolic genes such as *Pparγ* (Peroxisome proliferator-activated receptor gamma), *Pck1* (Phosphoenolpyruvate carboxykinase 1), and *G6pc* (Glucose-6-phosphatase catalytic subunit), which are critical for glucose and lipid metabolism [97,98,99]. This results in increased chromatin accessibility at these loci and aberrant upregulation of metabolic genes in offspring, making the offspring prone to glucose intolerance and insulin resistance [97]. Paternal psychosocial stress may have analogous effects: studies in mice indicate that chronic stress or trauma in males can trigger changes in sperm histone marks and chromatin accessibility at stress-related genes [94]. For example, stress-induced signaling through glucocorticoid and catecholamine pathways has been linked to altered H3K9 and H3K27 methylation in the male germ line, targeting genes involved in HPA axis regulation and synaptic plasticity [94]. While the precise histone modifications in sperm under paternal stress are still being mapped, these observations support the idea that in animal models, a father’s life experiences (dietary or traumatic) can epigenetically alter histone modifications in his germ cells, with measurable impacts on offspring phenotype. Another line of evidence comes from the experimental disruption of histone methylation in the male germ line. Overexpression of the histone H3 lysine 4 demethylase LSD1/KDM1A during mouse spermatogenesis (which erases activating H3K4 methylation marks) leads to profound developmental defects in descendants [100,101]. In this model, sperm from LSD1-overexpressing males show significantly reduced H3K4 methylation, resulting in aberrant activation of the zygotic genome and dysregulated gene expression in the early embryo. Strikingly, despite the transgene being present only in the first generation, the abnormalities in gene expression and phenotypes persist in the subsequent generation that never overexpressed LSD1 [100,101]. Chromatin profiling of these sperm confirmed specific losses of H3K4me3 at numerous genomic regions, including at promoters that remain aberrantly expressed in embryos, suggesting that sperm-borne H3K4me3 carries heritable information [100]. These findings (from controlled genetic mice models) provide evidence that perturbing a particular histone modification in germ cells can transmit an altered epigenetic state and phenotype to offspring transgenerationally.

Studies in other model organisms offer additional insight into the specific histone modifications implicated in mediating transgenerational epigenetic inheritance. Work in *C. elegans* has shown that histone marks, like H3K4me3 (associated with active genes), H3K9me3, and H3K27me3 (associated with gene repression), can be transmitted across multiple generations under certain conditions [102,103]. The balance between these modifications is finely regulated by specific enzymes. SET-2 (SET domain-containing protein 2), a methyltransferase, adds H3K4me3, which promotes active transcription, while SPR-5 (Suppressor of Presenilin Defects 5), a demethylase, removes H3K4me3 to reset epigenetic states between generations [104,105]. Disruption of this balance can lead to heritable epigenetic phenotypes: environmental stresses that increase H3K4me3 in *C. elegans* germ lines cause aberrant gene activation that persists for two to three generations, although these changes typically diminish over time, reflecting the relatively plastic nature of H3K4me3 marks [106]. By contrast, constitutive heterochromatic marks tend to be more stable across generations in *C. elegans*. H3K9me3, catalyzed by the MET-2 (Histone-lysine N-methyltransferase) enzyme, promotes persistent silencing of transposable elements and repetitive sequences, thereby safeguarding genomic stability in descendants [107,108]. Additionally, H3K27me3-based repressive domains can be partially retained in *C. elegans* embryos, allowing silenced gene states to be “remembered” in progeny [109]. These findings highlight that in *C. elegans*, histone modifications associated with heterochromatin formation (e.g., H3K9me3, H3K27me3) can confer stable, long-term epigenetic memory across generations, whereas active modifications, such as H3K4me3, are more transient, facilitating rapid responsiveness and epigenetic plasticity.

The insights gained from *C. elegans* provide valuable perspectives on the plasticity and stability of epigenetic memory, though significant differences exist between its germline reprogramming processes and those of mammals [110]. In mammals, germline reprogramming resets histone modifications, making stable transgenerational inheritance less likely compared to model organisms like *C. elegans*, where partial resetting allows certain modifications, such as H3K4me3, to persist across generations, facilitating the inheritance of environmental adaptations [110,111]. This developmental reprogramming substantially constrains the likelihood of stable histone modifications being transmitted from parent to offspring. In mice, some epigenetic memory can evade reprogramming (especially in the paternal lineage), but these exceptions are the minority and typically occur in specific contexts (such as imprinted genomic regions or experimentally induced scenarios) [100,101]. Humans present additional layers of complexity, characterized by extensive germline epigenetic reprogramming and numerous environmental and social influences throughout the longer lifespan [57,112]. To date, there is no definitive evidence that a traumatic exposure in a human ancestor leaves a unique, persistent histone modification that directly causes a phenotype in later generations. Instead, as repeatedly emphasized throughout this review, most putative cases of “transgenerational trauma” in humans are explained by intergenerational factors (e.g., stress during pregnancy affecting fetal development or traumatized parents influencing offspring through behavioral and social pathways) rather than the germline transmission of specific chromatin states. Thus, while histone modifications are compelling mechanistic candidates for nongenetic inheritance, their role in transmitting trauma-induced effects across generations remains speculative.

## 4. Non-Coding RNAs in Transgenerational Inheritance

Non-coding RNAs (ncRNAs) are emerging as key regulators of epigenetic mechanisms and have been implicated in mediating biological responses to environmental factors such as psychological trauma, diet, or exposure to toxins, including potential transmission to offspring [29,113,114,115,116,117]. Studies in animal models indicate that various ncRNAs—such as miRNAs, piRNAs, and tRFs—respond to environmental stressors, and these altered ncRNA profiles in germ cells may influence offspring phenotype [112,118]. However, the current evidence predominantly supports intergenerational inheritance rather than transgenerational effects in mammals [56]. Many ncRNA changes observed in the offspring of stressed parents do not persist into further generations in the absence of continued stress, suggesting they are often transient signals rather than permanently heritable traits [56].

sncRNAs are abundant in germ cells, and their expression can be dynamically regulated by an organism’s environment, providing a potential mechanistic link between parental environmental exposures and molecular changes in the gametes that can affect offspring behavior and physiology [49,93,116,119]. For example, in male mice subjected to stress, certain sperm miRNAs are differentially expressed. Among these, miR-375 modulates components of the HPA axis, and its stress-induced changes in sperm have been correlated with altered stress reactivity and adaptive capacity in offspring [118]. Similarly, miR-135 and miR-124—regulators of serotonin signaling pathways—are dysregulated in response to stress and have been associated with anxiety- and depression-like phenotypes, as well as differential responses to antidepressants [120,121].

Direct experimental evidence supports the role of sperm-borne RNAs in influencing offspring phenotypes [113,118,119]. When mouse oocytes were fertilized via intracytoplasmic sperm injection (ICSI) using sperm that had been experimentally depleted of RNAs, the resulting embryos showed impaired developmental potential, underscoring the importance of paternal RNAs in normal embryogenesis [122]. In another striking experiment, injecting sperm RNA from stressed male mice into normal zygotes was sufficient to induce some of the behavioral and metabolic traits of the stressed fathers in the resultant offspring [118,119]. These findings demonstrate that sperm RNAs can act as carriers of environmental information, functionally recapitulating aspects of the parental experience in progeny.

Emerging evidence suggests that multiple classes of ncRNAs may cooperate to mediate these effects. Paternal stress in rodents has been associated not only with changes in miRNAs but also tRFs and other sncRNAs in sperm [113,115]. Some of these sncRNAs target the regulation of developmental signaling pathways. For instance, Wnt/β-catenin and Notch signaling (crucial for neurodevelopment and synaptic plasticity) are influenced by sncRNAs that are altered in the sperm of stressed fathers [118,123,124]. Beyond neurodevelopmental pathways, sncRNAs—particularly tRNA fragments—have been implicated in metabolic regulation: paternal exposure to a high-fat diet or chronic stress can alter sperm tRF profiles, with downstream effects on metabolic gene regulation (such as *Igf2* and *Pparα* in offspring) [113,115]. In one study, male mice fed a high-fat diet exhibited specific changes in sperm tRF populations; their offspring showed perturbations in glucose and lipid metabolism accompanied by changes in hepatic gene expression, indicating an intergenerational metabolic dysregulation likely mediated by sperm ncRNAs [125]. Notably, these metabolic effects were clear in the F1 generation but tended to diminish in F2 if the high-fat diet was not maintained, highlighting that the persistence of ncRNA-mediated effects often requires reinforcement by continued environmental pressures [126].

Mechanistically, how might ncRNAs transmit information across generations despite the epigenetic reprogramming that occurs in early development? One possibility is through post-translational modifications of the ncRNAs themselves. For example, sperm sncRNAs can carry modifications (like methylation of RNA bases) that might enhance their stability or alter their function [113,116]. The enzyme DNMT2 (DNA (cytosine-5)-methyltransferase 2), which methylates small RNAs, has been shown to protect tRNA fragments from degradation, suggesting a mechanism by which certain stress-induced RNA molecules could survive longer and exert effects in the embryo [127]. However, direct evidence for ncRNA persistence beyond the F1 generation is sparse, in part because most epigenetic marks (including RNA modifications) are reset during germline and early embryonic development [128]. Another intriguing mechanism involves extracellular vesicles, such as exosomes [129,130,131,132]. There is evidence that stress in an organism can change the cargo of extracellular vesicles released into circulation, including the miRNAs they carry [133,134]. Extracellular vesicles can travel to the germ cells (sperm or oocytes) and deliver stress-responsive RNAs or other molecules, thereby conveying somatic stress signals to the germline [133,134]. Experiments in mice have shown that sperm incubated with exosomes collected from stress-treated epididymal epithelial cells produced offspring with altered neurodevelopment and adult stress reactivity [134]. While this soma-to-germline communication via exosomal RNAs is a plausible pathway, their persistence across multiple generations remains uncertain. In simpler organisms like *C. elegans*, small RNAs have been documented to transmit acquired traits (e.g., responses to starvation) across multiple generations, yet their influence diminishes over time, suggesting a potential decay of the epigenetic signal [135,136]. In mammals, sncRNA-mediated changes are most pronounced in the first-generation offspring (F1) and, in some cases, extend to the second generation (F2) when the parent is male, but they generally do not persist beyond that without continued exposure or reinforcement [71,137,138].

In summary, ncRNAs form an important part of the epigenetic response to trauma and stress. The diverse types of ncRNAs and their targets allow for a broad influence on developmental programs. However, as with DNA methylation, most documented ncRNA effects across generations in mammals are intergenerational and often require the initial trigger to be present in each generation to continue. The stability of these signals across true transgenerational timescales is an area of ongoing investigation. As advanced tools, like RNA sequencing and CRISPR-based editing of RNAs, become more sophisticated [139,140,141,142,143,144], researchers are starting to unravel the complexities of ncRNA-mediated inheritance and to discern which aspects might be adaptive, neutral, or maladaptive [71,112]. It appears increasingly likely that sncRNA transmission is a nuanced regulatory mechanism rather than a strict “memory” of ancestral environments—perhaps fine-tuning offspring development in anticipation of similar environments, but also readily overwritten by new experiences in subsequent generations [71,145,146,147].

## 5. Challenges and Controversies in Studying Trauma Across Generations

Investigating the effects of trauma within and across generations in humans presents numerous challenges spanning biological, environmental, ethical, and sociocultural domains [71,116,148]. Unlike laboratory animals, humans cannot be subjected to controlled trauma experiments for ethical reasons, so researchers must rely on observational and retrospective studies of naturalistic trauma (e.g., wars, disasters, familial abuse). This inherently limits the ability to determine causality, as human studies are often influenced by numerous confounding variables [71,149]. Environmental factors such as diet, socioeconomic status, education, social support, and concurrent life stressors can all influence epigenetic marks, making it difficult to attribute any observed epigenetic difference solely to an ancestral trauma [71,149]. Even in controlled animal models, the transmission of epigenetic changes beyond two or three generations is not consistently observed; many transgenerational effects attenuate over time or depend on specific environmental or biological conditions [150].

Another major challenge is the plasticity of epigenetic marks. The inherent plasticity of the epigenome, which enables it to adapt to environmental changes, also renders these epigenetic marks susceptible to resetting or loss [11,14]. In mammals, during germ cell development and early embryogenesis, a process of epigenetic reprogramming broadly resets genomic epigenetic marks—including DNA methylation, histone modifications, and non-coding RNAs—to achieve developmental totipotency [71,148,151,152,153]. During germ cell development, primordial germ cells undergo global DNA demethylation through active TET-mediated oxidation of methylated cytosines and passive dilution due to reduced DNMT1 activity, accompanied by widespread removal of repressive histone modifications (H3K9me3, H3K27me3) and dynamic shifts in non-coding RNA populations, particularly piRNAs, which re-establish methylation patterns on repetitive elements [27,92,115,128,154]. Following fertilization, early embryogenesis involves asymmetric epigenetic resetting: the paternal genome experiences rapid, active DNA demethylation coupled with the replacement of protamines by histones, while the maternal genome undergoes slower passive demethylation and histone remodeling, gradually establishing an open chromatin state essential for embryonic development [19,27,92,115,128,154].

This reprogramming is thought to safeguard development by clearing any potentially deleterious epigenetic alterations acquired during the parent’s lifetime [71,148,153]. While a small subset of loci (e.g., imprinted genes and, perhaps, some repetitive elements like LINE-1) can evade complete erasure, evidence for widespread escapee loci carrying over environmental information is limited [59,155,156]. In animal models, so-called metastable epialleles (MEs) are loci where the epigenetic state is variably set during development and can be influenced by the maternal environment (such as nutrition), leading to phenotypic variation among genetically identical individuals [155,157]. The Agouti viable yellow (*Avy*) mouse is a classic example: dietary factors in pregnant mice can alter the methylation of the *Avy* locus in their offspring, causing changes in coat color and metabolism that persist throughout life [158,159,160,161,162]. However, even these ME effects in mammals are generally considered intergenerational rather than transgenerational. In humans, there is evidence for metastable epialleles that respond to prenatal environmental conditions (for instance, loci that show methylation changes associated with seasonal maternal nutrition in certain populations), but whether these contribute to transgenerational trauma effects is not known [155,157,163,164].

The separation of the germline from somatic cells, known as the Weismann barrier, also complicates transgenerational epigenetic inheritance [165,166,167,168]. Somatic cells can undergo epigenetic changes in response to trauma, but those changes cannot directly influence the DNA methylation or chromatin state of germ cells due to this barrier [165,166,167,168]. For an epigenetic modification to be heritable, it must occur directly within the germ cells (sperm or oocytes) or be induced in these cells through a signaling mechanism originating from somatic cells. Studies are examining whether severe stress can bypass the Weismann barrier—perhaps through stress hormones or exosomal communication reaching germline cells [129,130,131,132,169]—but evidence in humans is lacking [150,168,170]. Thus, while intriguing examples of soma-to-germline influence have been proposed (as discussed with exosomes and ncRNAs), the paradigm in human biology is still that germline epigenetic reprogramming will erase most traces of a parent’s somatic experience.

In humans, the evidence for intergenerational epigenetic effects of trauma comes primarily from observational studies linking an ancestral exposure to some molecular or health outcome in descendants [65,66,171]. For instance, as mentioned, altered DNA methylation at stress-responsive genes (such as *NR3C1*, *FKBP5*, and *IGF2*) has been reported in children of individuals exposed to severe trauma, like the Holocaust or famine [65,66,171]. These findings indicate a correlation between parental stressors and epigenetic modifications in the next generation, suggesting that ancestral trauma might influence offspring stress physiology or behavior; however, this evidence is correlational and does not demonstrate a direct cause-and-effect relationship [64,65,94,130,131,135]. Disentangling inherited epigenetic changes from the effects of growing up with traumatized parents in a trauma-affected community is extremely difficult. This complexity arises because these environmental factors can independently or interactively influence offspring behavior and physiology, potentially mimicking or amplifying the effects attributed to epigenetic inheritance and leaving the causative role of epigenetics uncertain and unproven to date [55,154,155].

Moreover, even when trauma-induced epigenetic modifications (like DNA methylation changes in *FKBP5* or *NR3C1*) are observed in first-generation offspring, evidence of their persistence into further generations without renewed exposure is lacking [56,57,64]. In contrast with the more controlled and reproducible conditions of animal studies, where specific environmental stressors (e.g., controlled dietary or stress exposures) can be directly linked to epigenetic changes and phenotypic outcomes [49,51], human studies suggest these changes are often transient and highly context-dependent [112,172]. For instance, *NR3C1* methylation levels have been shown to be influenced by factors such as maternal behavior and early-life stress, highlighting the dynamic and reversible nature of these modifications [77,80,173]. This dynamic reversibility implies that the epigenetic state of one generation does not deterministically dictate the epigenetic landscape of the next; rather, the epigenome retains the capacity to recalibrate in response to the environmental conditions experienced by the offspring. Consequently, human studies have not yet been able to demonstrate multigenerational epigenetic effects independent of environmental and cultural factors [56,57,112].

An additional layer of complexity arises from the differential contributions of maternal and paternal factors to potential epigenetic inheritance [115,123,174,175]. Maternal trauma during pregnancy can directly affect the developing fetus (F1) and even that fetus’s germ cells (F2), as discussed, whereas paternal trauma is mediated through epigenetic changes in sperm [115,175,176]. Maternal trauma during pregnancy provides insights into potential intergenerational mechanisms, as gestational stress alters fetal development through changes in the maternal HPA axis [77,149,153,171,177]. Elevated maternal cortisol levels, a hallmark of HPA axis activation, can cross the placenta and influence fetal brain development by altering the expression of stress-related genes via DNA methylation and histone modifications [173,177]. These changes often target stress-related and immune-regulation genes, shaping offspring susceptibility to stress and immune dysfunction [171]. Additionally, disruptions in maternal oxytocin signaling, critical for caregiving behaviors, may impair maternal–infant bonding, further propagating the effects of trauma [178]. Maternal stress can also disrupt placental function, modulating the transfer of nutrients and hormones to the fetus, thereby affecting developmental trajectories [179]. These changes, while significant, are classified as intergenerational rather than transgenerational because they result from direct fetal exposure to maternal stress hormones rather than germline-mediated epigenetic inheritance (Figure 2).

The paternal contribution also suggests a complex interplay of factors [115,180]. For instance, alterations in ncRNAs, such as miRNAs and tRFs, have been observed in the sperm of stress-exposed males [113,115]. However, disentangling the biological effects of these RNA changes from environmental influences, such as stress-induced alterations in parenting behaviors, remains a significant challenge [115,180]. For example, parental trauma, such as PTSD in refugee parents or combat veterans, can significantly compromise family dynamics and children’s mental health through multiple, interrelated mechanisms [71,181,182]. Parents with PTSD often exhibit impaired parenting behaviors, characterized by emotional dysregulation, hypervigilance, or reduced sensitivity, that undermine children’s sense of security and contribute to emotional and behavioral difficulties [181]. Among displaced and refugee children resettled in high-income countries, these vulnerabilities are compounded by cultural displacement, discrimination, and socioeconomic hardship, further illustrating the intricate interplay of factors in the transmission of trauma [183]. Cross-fostering experiments in animal models, where offspring are raised by nonbiological parents, are commonly used to disentangle genetic and environmental influences on behavior and physiology. For example, studies have shown that the intragenerational impact of low maternal care on offspring phenotype and glucocorticoid receptor methylation can be reversed by cross-fostering, indicating that these effects are behaviorally mediated through maternal care rather than genetic/epigenetic inheritance [80,184]. In humans, analogous studies are challenging due to ethical and practical constraints.

It is also increasingly recognized that genetic predispositions and gene–environment interactions intersect with epigenetic factors in shaping trauma outcomes across generations [68,185,186]. Twin studies indicate a genetic component to PTSD susceptibility following trauma exposure [68,185,186]. Polymorphisms in genes like *FKBP5* and the serotonin transporter (*SLC6A4*) have been linked to increased susceptibility to trauma-related psychopathology, particularly when combined with environmental factors like early-life adversity or chronic stress [187,188]. For example, individuals carrying specific *FKBP5* risk alleles exhibit greater DNA demethylation in response to childhood abuse, leading to dysregulated HPA axis activity and heightened vulnerability to stress-related disorders such as PTSD and depression [189]. This underscores the intricate interplay between genetic and epigenetic factors, emphasizing the complexity of disentangling their respective contributions.

### Genetics/Epigenetics May Load the Gun, but the Environment Pulls the Trigger

Given the intricate interplay between biology and the environment, it has been suggested that genetic and epigenetic factors may prime individuals for certain outcomes, but the environmental context ultimately determines whether those outcomes manifest [187,190,191,192]. This perspective moves beyond a simple nature vs. nurture dichotomy and is encapsulated by the saying: “genetics/epigenetics may load the gun, but the environment pulls the trigger”. In the context of intragenerational trauma, this means that a descendant of trauma survivors might have a particular biological vulnerability (for example, epigenetic modifications in stress-regulatory genes like *NR3C1* that were influenced by intragenerational trauma) [187,190,191,192]. However, whether that vulnerability leads to a disorder, such as PTSD or depression, would depend on the descendant’s own life stressors and support systems. The diathesis–stress model in psychiatry offers a useful framework: it posits that individuals have varying levels of predisposition (diathesis) that require environmental stress to trigger pathology [187,191,192,193,194]. For example, trauma-induced epigenetic modifications, such as methylation of *NR3C1*, can influence stress regulation pathways and create predispositions for disorders like PTSD [191,192]. However, these epigenetic changes do not inevitably translate into psychopathology; their impact depends on the presence and severity of subsequent environmental challenges [191]. Supporting this, evidence shows that not all children of trauma survivors exhibit psychological distress, with many demonstrating resilience, particularly when raised in nurturing and supportive environments [195]. In contrast, individuals without a familial history of trauma can still develop PTSD and other stress-related disorders when exposed to extreme stressors, emphasizing the critical role of environmental factors in shaping psychological outcomes [196]. One implication is that improving the environment (for instance, through community support, therapy, or social policy changes) can effectively “put the safety on the gun,” preventing negative outcomes even in biologically at-risk individuals.

## 6. Pathways to Healing: Reversing the Epigenetic Effects of Trauma

Understanding that trauma’s impact is mediated by both biological and environmental factors opens the door to holistic healing approaches. A critical need exists for integrated, family-centered interventions to mitigate the cycle of trauma across generations. Addressing only the individual in isolation may be insufficient; instead, supporting the family system and broader community can create an environment where trauma’s effects are less likely to be passed on. For example, trauma-informed parenting programs have been shown to improve parents’ mental health and caregiving practices, which in turn can reduce the intergenerational transmission of stress and trauma-related behaviors to children [197]. These programs educate parents about the impact of trauma, teach skills for emotional regulation and responsive parenting, and help break harmful parenting cycles [197]. A meta-analysis of such programs indicates they can significantly decrease parents’ PTSD and depression symptoms while improving parent–child relationships [197].

In medical settings, family-centered care approaches have demonstrated benefits as well. For instance, in neonatal intensive care units (NICUs), involving and empowering parents in the care of their premature infants (e.g., through skin-to-skin contact, parental presence in rounds, and decision-making) alleviate parental stress and strengthens parent–infant bonding [198].

On the individual level, trauma-focused psychotherapies are a mainstay of treatment for PTSD and related conditions, and, intriguingly, there is emerging evidence that successful therapy can lead to measurable changes in epigenetic markers [199,200,201]. For instance, Cognitive Behavioral Therapy (CBT) for PTSD has been linked to modifications in *NR3C1* methylation in some studies, correlating with improved HPA axis regulation and symptom reduction [201,202]. Treating maternal depression during pregnancy—whether through psychotherapy (such as CBT or interpersonal therapy) or SSRIs (Selective Serotonin Reuptake Inhibitors)—can also alter epigenetic profiles in children [203]. In a pilot study, children whose mothers received treatment for depression showed different DNA methylation patterns at stress-related genes (*NR3C1*, *FKBP5*) compared to children of untreated depressed mothers, suggesting that effective treatment blunted the intergenerational epigenetic impact of maternal stress [203]. However, SSRIs, despite their potential to mitigate epigenetic changes linked to maternal stress, have been associated with risks when used during pregnancy, including neonatal adaptation syndrome, risk of persistent pulmonary hypertension of the newborn, and inconsistent evidence regarding long-term neurodevelopmental outcomes, such as autism spectrum disorder or attention deficit hyperactivity disorder (ADHD) [204,205,206,207,208,209,210]. Nonetheless, extensive systematic reviews indicate that after controlling for maternal psychiatric disorders, long-term prenatal antidepressant exposure was significantly associated only with elevated offspring body mass index (BMI) and an increased risk of developing affective disorders, whereas previously reported associations with autism spectrum disorders ADHD did not remain significant [210]. These findings indicate that previously observed associations between prenatal antidepressant exposure and adverse physical, neurodevelopmental, and psychiatric outcomes are largely attributable to confounding effects of underlying maternal psychiatric disorders [210]. Other therapies like Narrative Exposure Therapy (NET), which is often used for survivors of severe trauma, have been associated with epigenetic changes as well [199]. In conflict-affected populations, PTSD patients who responded to NET exhibited increased methylation at specific *NR3C1* CpG sites after therapy, and these changes were correlated with clinical improvement in PTSD symptoms [199]. Similarly, therapies such as Eye Movement Desensitization and Reprocessing (EMDR) and trauma-focused CBT have been shown, in some studies, to induce changes in DNA methylation of genes like *ZFP57* (Zinc Finger Protein 57), which is involved in genomic imprinting and maintaining epigenetic stability [202]. One longitudinal study of veterans with PTSD found that successful treatment not only alleviated symptoms but also reversed certain DNA methylation marks associated with the disorder [202]. Notably, *ZFP57* methylation changes were connected to both the development and remission of PTSD, suggesting that this epigenetic mark could potentially serve as a biomarker for treatment response [202].

Systematic reviews reinforce these findings by showing that therapy responders often exhibit specific changes in DNA methylation (for example, decreases in methylation at the *MAOA* (Monoamine Oxidase A) gene), which were not seen in non-responders [211,212]. These alterations not only correlate with symptom improvement but also suggest that epigenetic markers like *ZFP57*, *NR3C1*, and *MAOA* could serve as predictors of treatment efficacy, enabling more personalized approaches to trauma therapy [200,213,214,215]. However, caution is warranted as current evidence highlights significant limitations of these epigenetic markers, including variability across individual patients, dependency on the tissue sampled, and the dynamic influence of environmental and lifestyle factors [216,217]. Additionally, many markers identified thus far lack the specificity and replicability necessary for clinical use as standalone predictive tools, underscoring the need for extensive validation through large-scale, longitudinal, and multicenter studies before these biomarkers can be reliably implemented in personalized clinical practice [213,216,217]. The association between differential DNA methylation patterns and treatment response is an evolving area of research, representing a significant advancement toward personalized psychiatry [214].

It is, however, important to underline that while numerous therapies have been associated with epigenetic modifications, establishing a direct causal relationship between these changes and therapeutic outcomes remains a challenge [218]. In other words, the presence of an epigenetic shift following therapy does not necessarily confirm that the therapeutic effects are mediated by that specific epigenetic alteration.

Beyond individual psychotherapy, environmental enrichment and positive social experiences have shown substantial potential in mitigating trauma-induced epigenetic changes [219,220]. The concept of environmental enrichment involves providing a setting that is stimulating both mentally and socially—Mark Rosenzweig first described it as a combination of inanimate (physical) and social stimulation that enhances sensory, cognitive, and motor experiences [221]. In rodents, placing animals in enriched environments (with toys, opportunities for exercise, and social housing) after early-life stress, can normalize trauma-related modifications in DNA methylation and histone acetylation, particularly in genes linked to stress and emotional regulation, such as *Bdnf* and *Gad1* (Glutamate Decarboxylase 1) [222,223,224]. For example, early environmental enrichment in mice counteracts the effects of early-life stress by restoring Bdnf levels in the basolateral amygdala and normalizing glucocorticoid receptor translocation, improving neural plasticity and stress regulation [224].

Translating environmental enrichment to humans involves community and cultural interventions. Community-based programs that build familial cohesion, cultural reconnection, and socioeconomic empowerment can create the conditions for healing systemic and collective trauma [225,226,227]. For instance, programs in Indigenous communities that emphasize reconnecting with traditional culture, language, and land (often termed “connection to Country” in Australian Aboriginal contexts) have been associated with improved mental health and well-being [228,229]. There is evidence that Indigenous communities that have actively revived cultural practices and strengthened cultural identity have markedly lower youth suicide rates than those that have not [227].

By addressing social determinants of health and restoring a sense of identity and continuity, these interventions likely reduce chronic stress and enhance support systems, which, in turn, could modulate biological stress markers in the population [230,231]. They buffer against the transmission of trauma by changing the narrative from one of loss and suppression to one of pride and resilience. These community-level healing initiatives complement individual therapies by ensuring that the environment is conducive to healing and fostering resilience at a larger scale [230,231].

Innovative approaches like psychedelic-assisted therapies are also gaining attention for their potential to address trauma-related disorders by catalyzing profound psychological experiences and neural plasticity [232,233,234,235,236,237,238,239,240]. Classic psychedelics (such as psilocybin, Lysergic acid diethylamide (LSD), ayahuasca), and entactogens, like MDMA (3,4-Methylenedioxymethamphetamine), have demonstrated significant therapeutic potential in clinical trials for PTSD and depression, particularly when integrated into structured psychotherapeutic frameworks [232,233,234,235,236,237,238,239,240]. These substances appear to enhance emotional processing, facilitate the revisiting and reconsolidation of traumatic memories, and promote openness and empathy, which can be harnessed in therapy sessions [233]. From a neurobiological perspective, psychedelics induce transient states of hyperplasticity in the brain [237,238]. A recent study found that repeated doses of LSD in mice led to changes in methylation and the expression of genes involved in neural plasticity and stress response (including *Bdnf*), suggesting a direct epigenetic effect underlying its long-lasting effects on the brain [241]. Furthermore, a single high dose of a psychedelic in mice was shown to induce sustained increases in synaptic connectivity and histone acetylation at genes related to synaptic function, indicating that psychedelics can cause durable reprogramming of neural circuits at the epigenetic level [240].

Psilocybin-assisted therapy has been proposed as a means of “resurrecting ancestral familial health” by allowing families to process multigenerational grief and trauma together in an emotionally open state [239]. By realigning families with what developmental psychology calls the evolved “developmental niche” (environments of strong social support, ritual, and connection), such approaches aim to restore healthy patterns that support psychological and physiological well-being across generations [239]. Although these ideas are still theoretical, they suggest that beyond the individual patient, psychedelic therapies might be leveraged to heal relational and transgenerational wounds, potentially even influencing biomarkers of stress and resilience in family members. Indeed, an observational study on ceremonial ayahuasca use indicated changes in DNA methylation of the *SIGMAR1* gene (which encodes the sigma-1 receptor involved in neuroplasticity and stress responses) in participants correlated with improved emotional processing of trauma [242,243,244]. The sigma-1 receptor has been implicated in modulating epigenetic states during memory reconsolidation, suggesting that ayahuasca’s psychological effects could involve an epigenetic “unlocking” of traumatic memories and the laying down of new, healthier emotional narratives [242].

When combined with psychotherapy, psychedelic-assisted interventions address multiple dimensions—biological, psychological, social, and even spiritual [234]. They can produce a profound sense of unity, meaning, and emotional release that might be particularly suited to resolving the kind of deep-seated, often preverbal trauma passed down in families. By inducing a state of malleability in the brain, they may allow patients to rewrite ingrained patterns of thought and behavior, potentially reflected in epigenetic signatures, as noted above [234]. It is, however, important to underline that while early results are promising, more research is needed to determine the long-term effects of psychedelics on the epigenome and whether any such changes translate to transgenerational benefits. Regarding the risks associated with psychedelic therapies, evidence highlights potential acute psychological effects such as transient anxiety and perceptual disturbances, along with rare instances of Hallucinogen Persisting Perception Disorder (HPPD) and persistent psychosis [245,246]. Adverse outcomes are predominantly associated with unsupervised recreational use rather than structured therapeutic settings [245,247]. Clinical guidelines commonly exclude individuals with a family history of psychiatric conditions, including schizophrenia or bipolar disorder, to minimize risks of triggering psychotic episodes [247,248]. A recent comprehensive meta-analysis demonstrates that psychedelics are well-tolerated, with a low risk of emerging serious adverse events in a controlled setting with appropriate inclusion criteria [246].

Another category of healing pathways is mind–body interventions, such as meditation, mindfulness practices, yoga, and physical exercise. These interventions have well-documented benefits for reducing stress, anxiety, and PTSD symptoms [249,250,251,252,253], and research has begun to uncover associated epigenetic changes. Mindfulness-Based Stress Reduction (MBSR), a 9-week program teaching mindfulness meditation, has been shown to modulate stress responses and is associated with changes in DNA methylation of stress-related genes [212,254,255]. In particular, the practice of mindfulness or yoga has been linked with altered methylation of *FKBP5* and *SLC6A4* in peripheral blood cells [212,254,255]. These changes are thought to reflect reduced activation of stress pathways; for example, increased *FKBP5* methylation could enhance the negative feedback regulation of cortisol, thus reducing stress responses. A study on PTSD patients undergoing MBSR found that responders exhibited increased *FKBP5* methylation and cortisol normalization, whereas non-responders showed no such changes [212]. Similarly, veterans with PTSD who underwent meditation-based therapy showed that their pre-treatment *SLC6A4* genotype and methylation status predicted their response, underlining the interplay of genetics and epigenetics in responsiveness to mind–body therapy [254].

Beyond clinical populations, meditation has been found to correlate with epigenetic markers in healthy individuals as well. Novel research on Preksha Dhyāna, a form of mindfulness meditation, in college students found improvements in cognitive performance accompanied by differential DNA methylation in genes related to neural plasticity and stress regulation [256,257,258].

Regular physical activity is another powerful modifier of stress and mood that also has epigenetic effects. Numerous systematic reviews confirm that exercise interventions can significantly reduce symptoms of depression and anxiety [250,251]. Exercise has been shown to induce an increased global DNA methylation in skeletal muscle [259], leading to enhanced gene expression that supports metabolic adaptation [260], as well as changes in the expression of genes in the brain that are involved in mood regulation (such as *BDNF*) [261]. These effects have been shown to exert systemic benefits such as the modulation of inflammatory pathways, a reduction in pro-inflammatory markers, and an enhancement in anti-inflammatory responses, contributing to overall health and resilience [260]. Though the connection to trauma is indirect, a physically active lifestyle in a trauma survivor or in the next generation could improve overall resilience, reduce inflammation, and promote overall mental and physical health.

Overall, the integrative framework emerging from these findings is that interventions that foster a supportive environment (psychologically and physically) not only help immediate symptoms but can induce biological changes that promote resilience [211,223]. This integrative framework not only supports individual recovery but also fosters resilience and well-being across generations, offering a pathway to disrupt cycles of trauma and promote enduring health at both personal and collective levels [225,226,262].

## 7. Ethical Considerations in Framing Trauma as Inheritable

Framing trauma as an inheritable (biologically transmissible) condition carries profound ethical and social implications. On the one hand, recognizing a biological basis for intergenerational trauma can validate the experiences of affected communities by affirming that the struggles of descendants are not “just in their heads” but have real, embodied components. This perspective can foster compassion and inform targeted interventions [230,263]. On the other hand, there is a risk that emphasizing inheritance could inadvertently promote deterministic or fatalistic narratives that stigmatize already marginalized populations [230,263]. If misunderstood, it might imply that certain populations are biologically “damaged” or predisposed to dysfunction across generations, which could reinforce negative stereotypes or lead to discrimination. A major concern is that the discourse on epigenetic inheritance could divert attention from the social and structural causes of trauma that perpetuate trauma in these communities [225,230,263,264]. Historical traumas, such as colonization, forced assimilation, and systemic racism, have not only disrupted cultural continuity but also perpetuated cycles of socioeconomic and psychological distress [225,226,230]. This is particularly salient for Indigenous communities. As Gone et al. argue in the Australian Aboriginal context, research on intergenerational trauma must be grounded in the specific historical and cultural realities of Aboriginal peoples [230]. They emphasize that colonization, forced child removals (e.g., the Stolen Generations), and cultural suppression are the primary drivers of intergenerational trauma in these communities [230]. To attribute the ongoing disadvantages of Aboriginal and Torres Strait Islander peoples merely to “inherited trauma” without constant reference to these historical and structural factors could inadvertently support a narrative of inherent vulnerability or damage in Indigenous people, rather than a narrative focusing on resilience and the need for reparative justice [149,230]. These are exemplified by the ongoing impacts of the Stolen Generations, where policies of assimilation disrupted kinship systems and cultural continuity [228]. To avoid harm, it is essential for research in this area to employ culturally grounded methodologies in engaging with these communities. This means involving Indigenous scholars and elders, respecting Indigenous knowledge systems, and ensuring that research questions and interpretations are aligned with the communities’ values and needs [230,265,266]. Community-led healing initiatives, such as reconnection to land (“Country”), revitalization of cultural practices, and Indigenous-run trauma programs, should be highlighted as solutions, ensuring that the focus remains on healing and empowerment [230,265,266].

Another ethical aspect is the communication of research findings. Scientists and media must be careful not to oversimplify or misinterpret the scientific evidence. Headlines like “Epigenetics—do you have trauma in your genes? [267]” may capture attention, but they risk overlooking critical nuances and fostering unwarranted fear or misconceptions. Communities should hear about the plasticity of epigenetic mechanisms and the many pathways to resilience. In summary, by viewing epigenetic insights as one piece of a larger puzzle, we can avoid the pitfalls of determinism and instead use this knowledge to advocate for those who carry historical trauma.

## 8. Breaking Generational Chains: A Personal Reflection

From an evolutionary and historical perspective, trauma has been a near-universal experience of humanity: wars, forced migrations, enslavement, colonization, and social upheavals have left few cultures untouched. If trauma’s effects were purely and inescapably biologically inherited, one might expect humanity to be trapped in a downward spiral of ever-accumulating wounds. Yet history and anthropology show the opposite: many groups have demonstrated resilience, recovering from atrocities and building healthy societies. Strong family systems, rich cultural traditions, community solidarity, and rituals of healing have enabled people to overcome even the worst horrors. This observation highlights a fundamental point: while biology (including genetics and epigenetics) may influence the legacy of trauma, it does not dictate destiny. Current scientific evidence in humans for transgenerational epigenetic inheritance of trauma is still lacking. What is clear instead is that familial environments and broader socioeconomic and political factors are the predominant determinants of whether cycles of trauma continue or are broken. In other words, nature (biology) may load the gun, but nurture (environment) usually pulls the trigger—and importantly, nurture can also disable the trigger. Overemphasizing the biological inheritance of trauma risks framing it as an inevitable fate, which could inadvertently overshadow the very real, addressable factors that perpetuate trauma across generations: systemic racism, poverty, injustice, and lack of access to care. It is these structural factors, combined with familial patterns of behavior, that largely decide whether a child of a trauma survivor will thrive or struggle. Therefore, the focus must shift to modifiable influences. We should channel our efforts into fostering supportive environments for children—environments with safe, stable, nurturing relationships. We must support revitalizing cultural practices and strengthening community connections — particularly among groups whose heritage has been systematically undermined — recognizing that culture serves as a vital source of resilience, identity and collective healing. In acknowledging the reality of intergenerational trauma, we must equally acknowledge intergenerational resilience: the ability of love, support and cultural continuity to interrupt cycles of harm and foster healing across generations. Our ancestors’ suffering may shape us, but with knowledge, compassion, and action, we can ensure that it is our choices and care in the present that shape the generations to come.

## 9. Research Directions

To advance the studies on trauma across generations, future research must adopt rigorous longitudinal, multigenerational study designs that account for environmental, sociocultural, and genetic confounders. Establishing baseline epigenetic states prior to stress exposure is critical to distinguish trauma-induced modifications from other contributing factors in transgenerational inheritance [52,268]. However, while longitudinal cohort studies are invaluable for identifying generational effects, their observational nature and susceptibility to confounding variables limit their ability to establish causality [71,149].

Advances in sequencing technologies and epigenome-wide association studies (EWAS) offer promising tools for disentangling the complex interactions between genetic, epigenetic, and socio-environmental factors [194,269]. For example, EWAS has been used to identify differential DNA methylation patterns associated with stress-related outcomes, broadening our understanding of trauma’s molecular signatures [149,270]. Furthermore, emerging methodologies such as high-throughput nanopore sequencing offer a groundbreaking approach to mapping DNA methylation, enabling high-resolution analysis of epigenetic modifications while preserving native DNA structures [269]. This could identify patterns (like allele-specific methylation or comethylation across distant sites) that might be missed by array-based methods.

Despite these advancements, challenges such as variability in study designs, population heterogeneity, and inconsistent methodologies hinder the development of a unified theoretical framework [271,272,273]. Addressing these issues requires methodological rigor and an interdisciplinary approach, integrating epigenomic profiling with behavioral analyses to distinguish inherited epigenetic effects from shared environmental influence [71,223,269]. Researchers must ensure that study designs are robust and reproducible, incorporating diverse populations to account for variability in genetic and environmental contexts [271,272]. Furthermore, collaboration between biological, social, and cultural disciplines is essential to create a holistic understanding of trauma and its effects across generations.

An important direction is to work closely with communities that have experienced historical trauma in participatory research frameworks. Partnerships with Indigenous and other marginalized communities can help ensure that study designs are culturally appropriate and address community-relevant questions [265,266,274]. Such collaborations (for example, involving community advisory boards or co-researchers) have shown value in integrating traditional knowledge with scientific inquiry [265,266,274]. They also help to interpret findings in culturally meaningful ways and to design interventions that the community is likely to embrace [265,266]. Indigenous concepts of wellness, which emphasize balance, connection to land, and spirituality, can guide the interpretation of how trauma and healing manifest biologically. Moreover, ethical concerns (like those discussed in the previous section) can be better navigated when the community has a say in how findings are communicated and used.

Another frontier is exploring interventions during critical windows. Pregnancy and early childhood are sensitive periods when interventions might prevent the solidification of trauma’s effects [275,276]. This will pave the way for evidence-based interventions tailored to interrupt the effects of trauma across generations.

## 10. Conclusions

The effects of trauma across generations represent a multifaceted challenge that intertwines biological, environmental, and cultural dimensions. While significant advances have explored the role of epigenetic mechanisms in encoding trauma’s impact, their applicability to human contexts remains constrained by methodological limitations and ethical complexities. To date, clear evidence of deterministic transgenerational epigenetic inheritance in humans is lacking; instead, the intergenerational transmission of trauma appears to be driven predominantly by sociocultural factors. Emerging therapies, including psychedelic-assisted psychotherapy and mindfulness-based interventions, offer innovative approaches to potentially reverse trauma’s biological and psychological imprint. The path forward requires interdisciplinary collaboration and culturally sensitive research methodologies. By integrating genomic tools with social science insights, and partnering with communities in research and healing efforts, we can develop a more holistic understanding of trauma’s legacy. Such an approach ensures that we address not only the molecular aspects of trauma but also the structural and relational determinants that allow trauma to persist. Ultimately, by uniting scientific innovation with community-led practice, we can foster resilience, disrupt the cycles of trauma, and create pathways for intergenerational healing and recovery. Each generation, armed with greater knowledge and better support, can move further from the shadows of the past toward a healthier, more hopeful future.

## Figures and Tables

**Figure 1 ijms-26-03075-f001:**
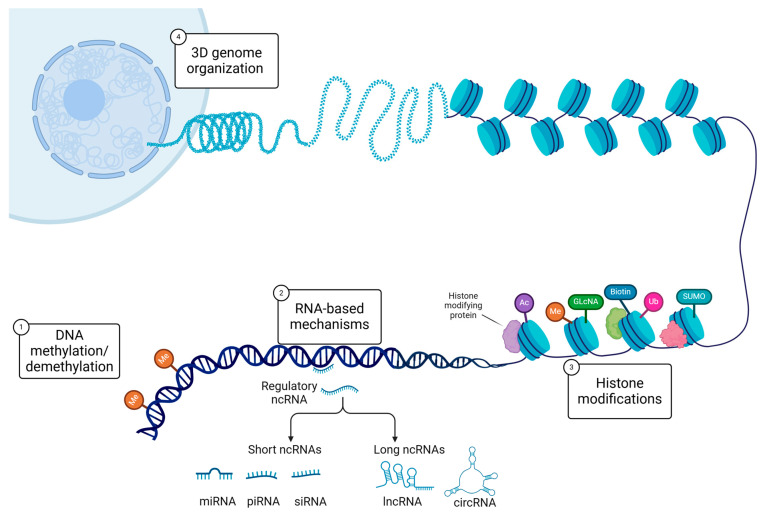
Overview of key epigenetic mechanisms in gene regulation. (**1**) DNA methylation and demethylation: the enzymatic addition or removal of methyl groups (–CH_3_) at cytosine residues (typically at CpG dinucleotides). DNA methylation is commonly associated with transcriptional repression, while demethylation can enable transcriptional activation. (**2**) RNA-based mechanisms: regulatory ncRNAs, including short ncRNAs (miRNAs, piRNAs, siRNAs) and long ncRNAs (lncRNAs, circular RNAs), mediate chromatin remodeling, transcriptional silencing, and mRNA degradation. (**3**) Histone modifications: post-translational modifications (acetylation, methylation, ubiquitination, SUMOylation) are added to histone proteins, modulating chromatin compaction and gene expression. These modifications create a dynamic “histone code” that can activate or repress transcription in a context-dependent manner. (**4**) Three-dimensional (3D) genome organization: the spatial organization of chromatin within the nucleus, including chromatin loops and topologically associated domains (TADs), which facilitate or inhibit interactions between regulatory elements and gene promoters, thus regulating transcription. Generated using Biorender, https://biorender.com/ (accessed on 24 March 2025).

**Figure 2 ijms-26-03075-f002:**
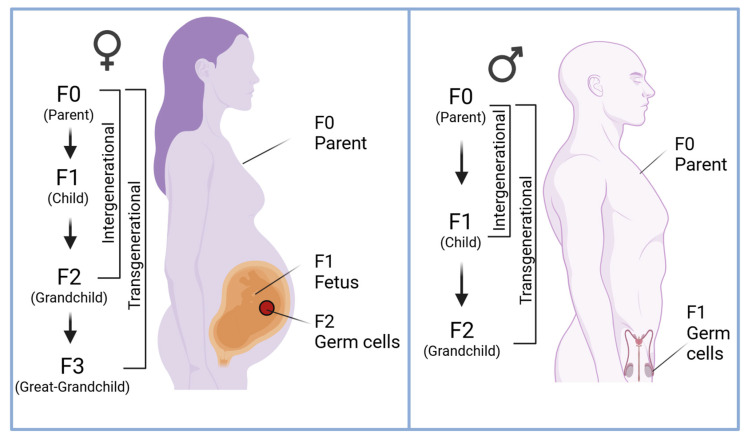
Intergenerational vs. transgenerational epigenetic inheritance in maternal and paternal lineages. In the maternal lineage (left side), an environmental influence on a pregnant F0 female can induce epigenetic modifications that affect her directly and are also present in the developing F1 fetus and the germ cells inside that fetus (which will give rise to the F2 generation). This means that both the F1 and F2 generations have direct exposure to the F0 trauma or stressor (through the in utero environment). Such effects observed in F1 (and F2, for maternal exposures) are classified as intergenerational inheritance since the affected generations were directly exposed. If an epigenetic modification persists into the F3 generation (the first generation not directly exposed in the maternal line), it represents transgenerational inheritance. In the paternal lineage (right side), an F0 male’s exposure can directly affect his sperm, thereby influencing the F1 generation. The F2 generation in the paternal line would be the first not directly exposed. Therefore, epigenetic changes appearing in F2 (or beyond) of the paternal line are considered transgenerational inheritance. In both scenarios, establishing definitive transgenerational epigenetic inheritance requires the exclusion of confounding factors associated with ongoing environmental exposures and the demonstration that epigenetic modifications are directly inherited via the germline. Generated using Biorender, https://biorender.com/ (accessed on 24 March 2025).

## Data Availability

Data sharing is not applicable.

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
