# Peer review of "Epigenetic Echoes: Bridging Nature, Nurture, and Healing Across Generations"

_ijms, 2025, doi:10.3390/ijms26073075_

Round 1

Reviewer 1 Report

Comments and Suggestions for Authors

Authort should more explicitly state that the evidence for transgenerational epigenetic inheritance in humans is still inconclusive and that most observed effects are likely intergenerational rather than truly transgenerational.

The author discusses maternal stress during pregnancy affecting the fetus (F1) and germ cells (F2) as if it were transgenerational, but this is actually intergenerational because the F1 and F2 generations are directly exposed. So distinction between intergenerational (direct exposure) and transgenerational (no direct exposure) effects should be more clearly emphasized, especially when discussing human studies.

The author acknowledges the Holocaust survivors and their descendants, attributing observed epigenetic changes to trauma inheritance, but does not fully account for the impact of shared trauma narratives, parenting styles, and cultural factors. So author should more explicitly state that shared environmental and cultural factors are often the primary drivers of intergenerational trauma, and epigenetic changes may be secondary or correlational rather than causal.

The author frequently cites animal studies to support claims about epigenetic inheritance in humans. The limitations of extrapolating animal findings to humans should be more clearly highlighted, especially when discussing transgenerational epigenetic inheritance.

The author focused heavily on epigenetic mechanisms but does not adequately address the role of genetic factors in trauma susceptibility and transmission. Author should more explicitly discuss the interplay between genetic and epigenetic factors in shaping trauma outcomes, emphasizing that epigenetics is just one piece of a larger puzzle.

The author discusses that epigenetic markers NR3C1, FKBP5, ZFP57 could serve as biomarkers for trauma and treatment response. However, the variability and context-dependence of these markers make them unreliable as standalone diagnostic tools. So author should add caution against overinterpreting epigenetic biomarkers and emphasize the need for more rigorous validation before they can be used in clinical settings.

The author discuss emerging therapies as if they can directly reverse epigenetic changes caused by trauma. However, the evidence for such effects is preliminary and largely correlational. So author should more clearly state that while these therapies show promise, their epigenetic effects are not yet fully understood, and more research is needed to establish causality.

Author should more thoroughly discuss the ethical risks of overemphasizing epigenetic inheritance and advocate for a balanced approach that considers both biological and socio-environmental factors.

The author did not sufficiently address the extensive epigenetic reprogramming that occurs during germ cell development and early embryogenesis, which erases most epigenetic marks.

The author should more explicitly link epigenetic research to broader socio-historical contexts, emphasizing that trauma is not just a biological phenomenon but also a cultural and historical one.

The author does not adequately address the variability in epigenetic responses to trauma.

The author focuses on germline epigenetic inheritance but does not sufficiently discuss the role of somatic epigenetic changes in trauma responses.

Reviewer 2 Report

Comments and Suggestions for Authors

This review examines the epigenetic mechanisms that have been correlated with inherited effects of trauma, including DNA methylation, histone modification, and non-coding RNAs. While it summarizes all three of these epigenetic mechanisms in the introduction, it only describes the role of DNA methylation and non-coding RNAs in inherited trauma. It also covers the challenges and controversies of these types of studies, as well as ways to reverse epigenetic effects of trauma and the ethical considerations in framing trauma as inheritable. Though at times the negative side effects of potential therapies are ignored, the review is unique in that it covers both the natural and social science aspects of inherited trauma. I recommend it for publication after addressing the following comments.

  • Histone modifications (along with DNA methylation and non-coding RNAs) are included in the introduction (lines 114-122) as an epigenetic mechanism that could lead to inherited effects of trauma. However, only DNA methylation (section 2) and non-coding RNAs (section 3) are discussed. A section on histone modifications should be added.
  • In Line 531 the authors state that “Scientists and media must be careful not to oversimplify or misinterpret the scientific evidence.” However, when talking about potential therapies, the authors themselves are guilty of this. Data is presented about how taking SSRI’s when pregnant can change the epigenetic profiles of offspring (Line 421). It is concluded that this has the positive effect of blunting the intergenerational epigenetic impact of maternal stress. However, there are many other potential negative side effects of taking an SSRI when pregnant and these are not even alluded to. Similarly, when discussing psychedelic therapies, they speak of how these therapies can “cause durable reprogramming of neural circuits at the epigenetic level" (line 468). Here it is implied that this is a beneficial result, but there are also many harmful effects that can be associated with this reprogramming, and these are not even mentioned. Lack of discussion of negative points can lead to dangerous misinterpretation of data.

Author Response

Comment 1: 

This review examines the epigenetic mechanisms that have been correlated with inherited effects of trauma, including DNA methylation, histone modification, and non-coding RNAs. While it summarizes all three of these epigenetic mechanisms in the introduction, it only describes the role of DNA methylation and non-coding RNAs in inherited trauma. It also covers the challenges and controversies of these types of studies, as well as ways to reverse epigenetic effects of trauma and the ethical considerations in framing trauma as inheritable. Though at times the negative side effects of potential therapies are ignored, the review is unique in that it covers both the natural and social science aspects of inherited trauma. I recommend it for publication after addressing the following comments.

Response 1: 

Thank you very much to the reviewer for the positive feedback and insightful comments. We have carefully addressed all the points raised, including expanding the section on histone modifications and ensuring that potential negative side effects of therapies are appropriately acknowledged in the manuscript.

Comment 2: Histone modifications (along with DNA methylation and non-coding RNAs) are included in the introduction (lines 114-122) as an epigenetic mechanism that could lead to inherited effects of trauma. However, only DNA methylation (section 2) and non-coding RNAs (section 3) are discussed. A section on histone modifications should be added.

Response 2:  We would like to thank the reviewer for this valuable comment. The section on histone modifications was originally part of the manuscript but was accidentally removed during revisions. We have now re-incorporated it into the manuscript as suggested: Section 3 (lines 164-226).

Comment 3: In Line 531 the authors state that “Scientists and media must be careful not to oversimplify or misinterpret the scientific evidence.” However, when talking about potential therapies, the authors themselves are guilty of this. Data is presented about how taking SSRI’s when pregnant can change the epigenetic profiles of offspring (Line 421). It is concluded that this has the positive effect of blunting the intergenerational epigenetic impact of maternal stress. However, there are many other potential negative side effects of taking an SSRI when pregnant and these are not even alluded to. Similarly, when discussing psychedelic therapies, they speak of how these therapies can “cause durable reprogramming of neural circuits at the epigenetic level" (line 468). Here it is implied that this is a beneficial result, but there are also many harmful effects that can be associated with this reprogramming, and these are not even mentioned. Lack of discussion of negative points can lead to dangerous misinterpretation of data.

Response 3: 
We thank the reviewer for raising important points regarding the potential risks associated with SSRIs in pregnancy and psychedelic therapies. Currently, evidence directly linking the discussed epigenetic mechanisms to these specific clinical risks is lacking. Nonetheless, to provide a balanced perspective, we have expanded the manuscript to discuss these safety considerations in Section 6 (lines 406-414 and 426-430).

Documented concerns of SSRIs during pregnancy include neonatal adaptation syndrome, a risk of persistent pulmonary hypertension, and conflicting data on neurodevelopmental outcomes in offspring. However, systematic reviews indicate that the long-term significance of these findings is considerably attenuated when controlling for the effects of maternal depression itself.

For psychedelic therapies, we have added an evidence-based discussion of risks associated with psychedelics, including acute psychological effects such as transient anxiety and perceptual disturbances, along with rare instances of Hallucinogen Persisting Perception Disorder (HPPD) and persistent psychosis. Additionally, we have reported data from a comprehensive meta-analysis demonstrates that psychedelics are well-tolerated, with a low risk of emerging serious adverse events in a controlled setting with appropriate inclusion criteria.

When discussing the long-term use of psychedelics is also important to highlight the historical context in which psychedelic substances have been safely utilized within structured ceremonial and traditional settings across many cultures for centuries, further underscoring the importance of appropriate context in their therapeutic application.

These revisions ensure that the review provides a comprehensive and nuanced appraisal of these interventions consistent with current scientific evidence and clinical consensus, directly addressing the reviewer's valuable feedback.

Round 2

Reviewer 1 Report

Comments and Suggestions for Authors

Thanks for revising the manuscripts.